# Bistable Threshold Humidity Sensor Switch with Rectangular Bimorph Bending Plate

**DOI:** 10.3390/mi11060569

**Published:** 2020-06-03

**Authors:** Nikolai Gulnizkij, Gerald Gerlach

**Affiliations:** Solid-State Electronics Laboratory, Technische Universität Dresden, 01062 Dresden, Germany

**Keywords:** sensor switch, switching hysteresis, threshold sensor, bistable sensor, plate theory, bimorph effect, relative humidity sensing

## Abstract

Energy-autonomous bistable threshold sensor switches have the potential to reduce costs because they do not need any electrical energy supply for monitoring physical quantities, such as relative humidity. In previous work, a bistable beam-like sensor switch with switching hysteresis was manufactured from sheet metal and a partially coated water vapor-sensitive hydrogel (poly(vinyl alcohol)/poly(acryl acid)). Based on the beam theory, a corresponding mechanical model was developed. However, bending plates should be used instead of bending beams to separate the humidity to be measured from the electrical contacts. For this reason, this work deals with the development and realization of a mechanical model based on the plate theory to describe the deflection of a silicon bimorph bending plate partially coated with hydrogel that swells with increasing humidity. For implementing a switching hysteresis a plasma-enhanced chemical vapor deposition silicon dioxide (SiO_2_) layer is used, which was deposited and structured on top of the silicon plate. The hydrogel layer itself is patterned on the surface of the bending plate using a stamp technique. To validate the mechanical model, the switching hysteresis of the miniaturized sensor switch was measured optically by a camera measurement device.

## 1. Introduction

The demand on sensors and sensor systems that combine data acquisition and signal processing increases in several fields, like industrial production, building automation, and automobile technology. For inline process observation and control, the most important quantities are temperature and humidity. Numerous sensors and sensor systems can be used for humidity sensing [1,2]. In building monitoring, ca. 90% of all sensors are used as threshold switches and, in process control, some 70% [3]. A novel approach for a non-powered humidity threshold switch based on the binary zero-power sensor principle (BIZEPS) was proposed in Reference [4,5,6,7,8,9,10]. With this principle, the energy for the switching process is taken directly from the measured variable. The basic part is a humidity-dependent polymer film in which swelling has both a sensory effect and an actuator function that causes switching.

Figure 1 shows the general set-up and working principle of a hydrogel-based humidity threshold switch. The silicon-based bending plate is located in the middle of a silicon die. The surface of the electrical contacts for the switch have to be level and flat. For this reason, a rigid center in the middle of the bending plate, a so-called boss structure, is used. The rigid center of the bending plate is deflected by the humidity-dependent swelling behavior of the hydrogel deposited on the silicon plate. The water vapor uptake, and hence the swelling state, depend on the humidity of the surrounding air. When swelling, the hydrogel provides the mechanical energy to deflect the bending plate without any electrical energy supply. This sensor shows a high sensitivity with regard to small changes and a corresponding closing (when swelling) or opening (when deswelling) of the microcontact [7]. 

## 2. Previous Work

In Reference [11,12] a hydrogel-based piezoresistive chemical sensor for pH measurement and in Reference [13] a hydrogel based biochemical sensor were introduced. To increase the applicability of the hydrogel-based sensors, the hydrogels were modified via γ–irradiation for biochemical applications [14] and also used for the detection of heavy metal ions [15]. For polymer modifications, nanocomposites were realized in Reference [16], and the influence of ion-implementation was investigated in Reference [17].

The nonlinear effects resulting from the swelling behavior of the hydrogel were modeled for pH-sensors [18] and biochemical sensors [19]. 

In Reference [20], the swelling behavior of a swellable hydrogel was used for flow control in microfluidic systems and, in Reference [21,22], as a sensor-actor system for monitoring and control of physical values (temperature). 

Due to the wide range of applications of hydrogel based sensor-actuator systems, there is a great demand on the development of a simple mechanical model for calculation of the actuator behavior or for the derivation of guidelines for the design of such sensor-actuator systems. Hydrogel-based sensor-actuator systems consist of a bending plate and a hydrogel manufactured for the respective application. From this, the plate theory can be used to develop a mechanical model to calculate the bending plate deflection in dependence of the hydrogel swelling.

In Reference [23], a beam-like structure was considered comprising a sheet metal-based double-sided clamped beam and a partial coating with hydrogel. For the calculation of the deflection of the bending beam as a function of the hydrogel coating, a model was developed by means of the beam theory and lumped mechanical models for the bending beam, where the maximum deflection occurs at a coverage ratio of ca. 50%.

To avoid both damages due to electrical arcing and oscillations at humidity values near the threshold, the humidity-deflection relationship should show an S-shaped hysteresis. Such a hysteresis—in conjunction with a corresponding pre-deflection of the bending beam—can easily be achieved by an appropriate axial compressive force. In Reference [23], the lumped model was extended by additional axial forces, acting from the two clamping points. It could be shown that the switching hysteresis (threshold value and hysteresis width) is adjustable via variation of the geometry parameters, like thickness and length of the bending plate. 

In addition to the requirements considered so far, other objectives should be taken into account for future solutions: To separate the electrical contacts from the measured humidity a bimorph bending plate should be used instead of a bending beam considered in Reference [23].Future sensors should be low-cost and miniaturized. For that reason, MEMS technologies should be preferred.

The latter can be easily realized by using silicon bending plates with boss structures in the center that provide the level and flat contact areas (see Figure 1). As already proposed in Reference [24], both a pre-deflection and an axial compressive force can be achieved by a silicon dioxide (SiO_2_) layer deposited also partially on the bending plate. Since SiO_2_ has a much lower coefficient of thermal expansion than silicon and is deposited at temperatures of several 100 °C, cooling down to room temperature leads to a larger expansion of the SiO_2_ thin film than of the silicon (Si) plate [8]. To model the mechanical switching behavior of the rectangular bimorph bending plates, more advanced models have to be applied in comparison with the simplified lumped models for bending beams, like in Reference [24,25,26]. The goal of the following considerations is to derive analytical models for the description of rectangular bimorph plates with patterned layers that undergo a volume expansion (swelling) due to humidity as similar to the thermal expansion of any materials. 

The paper is organized in the following manner. First, we will consider a homogenous, rectangular, shear-rigid-bending plate with respect to its general deflection equation and the occurring cutting moments and forces. In a second step, when a part of the plate is subjected to forces in the normal direction, this leads to internal bending moments around the neutral fiber of the plate. This corresponds to the effect of a bimorph element with one of the layers swelling under the influence of humidity. This analogy can easily be used to model the bimorph structure. It will be shown that analytical approaches, known from load-bearing shell structures, can be applied to derive analytical expressions of the deflection-load and deflection-humidity relationship. 

In a third part, the fabrication of a silicon-based MEMS threshold sensor is described, where the pre-deflection, as well as the axial compression pre-stress, is provided by a silicon oxide island in the middle of the bending plate. Experiments with such sensor specimens have shown that the patterned SiO_2_ layers on top of the silicon sensor chips are suitable to achieve such a targeted S-shaped hysteresis course. 

## 3. Plate Equations for Thin Rectangular Bimorph Plates

To derive an analytical equation for a bimorph bending plate with hysteretic behavior due to an applied compression axial force, the following procedure is used. First, the plate is considered as thin plate, which allows several simplifications (Section 3.1). In a second step, the equilibrium equations for the forces and moments are considered (Section 3.2) and—by including the laws of matter (Section 3.3)—the plate Equations are derived (Section 3.4). Using the analogy between the occurrence of an internal bending moment due to a bimorph plate structure and due to a locally applied area force, an analytical relationship for the deflection of a partially as bimorph acting structure can be derived (Section 3.5). In a last step, the influence of an additionally applied in-plane force is introduced to consider the hysteretic behavior (Section 3.6) as targeted in this application. 

### 3.1. Assumptions

To describe the deflection of a rectangular bending plate partly covered with hydrogel, several assumptions have been taken into account (Figure 2): Small deflections *w* in comparison to the plate thickness *h*: *w/h* < 0.2.The plate is thin, i.e., its thickness *h* be significantly smaller than the sizes $l_{1}$ and $l_{2}$ of its edges: *h*/$l_{1,2}$ < 0.1.Hooke’s law should apply.The plate is considered to have a large bending stiffness, i.e., it is flexurally rigid. The plate material is isotropic and homogeneous.The strain $\varepsilon_{33}$ in $x_{3}$-direction is negligibly small, so that the plate can be considered rigid along the thickness: $\varepsilon_{33}=0$The normal to the undeformed surface remains normal to the surface when the plate is deformed (Figure 2c). This property is similar to the assumption of a thin plate.

To derive a simplified mechanical model from the plate theory, only small curvatures of the plate center surface, are considered (Figure 3). It follows:The shear strains $\gamma_{13}$ and $\gamma_{23}$ are neglectable. Bernoulli’s hypothesis applies saying that the course of the stresses $\sigma_{11}$ and $\sigma_{22}$, as well as of the strains $\varepsilon_{11}$ and $\varepsilon_{22}$, change linearly along the thickness, whereas the shear stresses $\tau_{13}$ and $\tau_{23}$ show a parabolic course along h (Figure 2d) [27] (pp. 10–17).

For the given assumptions, the curvatures $\kappa_{11}$ and $\kappa_{22}$ of the central surface for the $x_{1}$*,*$x_{3}$- and $x_{2}$,$x_{3}$-planes can be linearized for small angles (Figure 3) [26,27]:(1)$$\kappa_{11}=\frac{-w_{,11}}{{\left[1+{\left(w_{,1}\right)}^{2}\right]}^{\frac{3}{2}}}\approx -\frac{d^{2}w}{dx_{1}^{2}}\approx -w_{,11},$$
(2)$$\kappa_{22}=\frac{-w_{,22}}{{\left[1+{\left(w_{,2}\right)}^{2}\right]}^{\frac{3}{2}}}\approx -\frac{d^{2}w}{dx_{2}^{2}}\approx -w_{,22}.$$

To derive the plate equation, the kinematic equations, the equilibrium conditions, and the law of matter are to be formulated with respect to the pre-introduced assumptions.

Three types of forces can act on the plate:forces perpendicular to the plate surface (out-of-plane),in-plane forces caused from the bimorph effect due to a coating of the plate, andin-plane forces caused by in-plane loads from the clamping.

For small deflections according to Figure 3, it yields:(3)$$\cos\varphi_{1}\approx \varphi_{1}\approx 1,$$
(4)$$\sin\varphi_{1}\approx \varphi_{1}\approx \tan\varphi_{1}=w_{,1},$$
(5)$$\sin\varphi_{2}\approx \varphi_{2}\approx \tan\varphi_{2}=w_{,2}.$$

The displacements $u_{1}$ in $x_{1}$-direction and $u_{2}$ in $x_{2}$-direction depend on $\varphi_{1}$ and $\varphi_{2}$ and, hence, on $w_{,1}$ and $w_{,2}$:(6)$$u_{1}\left(x_{1},x_{2},x_{3}\right)=-x_{3}w_{,1}\left(x_{1},x_{2}\right),$$
(7)$$u_{2}\left(x_{1},x_{2},x_{3}\right)=-x_{3}w_{,2}\left(x_{1},x_{2}\right),$$
(8)$$w\left(x_{1},x_{2},x_{3}\right)=w\left(x_{1},x_{2}\right).$$

This leads to the strain-displacement equations:(9)$$\varepsilon_{11}=u_{1,1}=-x_{3}w_{,11}=x_{3}\kappa_{11},$$
(10)$$\varepsilon_{22}=u_{2,2}=-x_{3}w_{,22}=x_{3}\kappa_{22},$$
(11)$$\gamma_{12}=u_{1,2}+u_{2,1}=-2x_{3}w_{,12}=2x_{3}\kappa_{12},$$
(12)$$\gamma_{21}=u_{2,1}+u_{1,2}=-2x_{3}w_{,21}=2x_{3}\kappa_{21}.$$

Since the functions for the normal strains $\varepsilon_{11}$, $\varepsilon_{22}$ and shear strains $\gamma_{12}$ = $\;\gamma_{21}$ are linear, they can be represented by the derivative of the deflection $w\left(x_{1},x_{2}\right)$ in $x_{3}$-direction [27].

### 3.2. Equilibrium Conditions

Since so far no axial forces occur in the plate, the normal stresses $n_{11}$, $n_{22}$, and $n_{33}$ arise from the stresses $\sigma_{11}$, $\sigma_{22}$, and $\sigma_{33}$, corresponding to the three axes of the coordinate system: (13)$$n_{11}=\int_{-h/2}^{h/2}\sigma_{11}dz=0,\;\;n_{22}=\int_{-h/2}^{h/2}\sigma_{22}dz=0,\;\;n_{33}=\int_{-h/2}^{h/2}\sigma_{33}dz=0.$$

Figure 4 and Table 1 show the acting length-related forces and moments at a differential part of the bending plate. Here, *q* and *m* are the length-related out-of-plate forces and moments, respectively, at an area $\Delta A=dx_{1}dx_{2}$. To derive the bending moments, the torsional moments and the transverse forces, the corresponding quantity is integrated via the corresponding mechanical stress along the thickness *h*.

In the equilibrium state, the following equations apply for the forces and moments: (15)$$-q_{1}dx_{2}+\left(q_{1}+q_{1,1}dx_{1}\right)dx_{2}-q_{2}dx_{1}+\left(q_{2}+q_{2,2}dx_{2}\right)dx_{1}+qdx_{1}dx_{2}=0,$$
(16)$$m_{22}dx_{1}-\left(m_{22}+m_{22,2}dx_{2}\right)dx_{1}+m_{12}dx_{2}-\left(m_{12}+m_{12,2}dx_{1}\right)dx_{2}+q_{2}dx_{1}\frac{dx_{2}}{2}+\left(q_{2}+q_{2,2}dx_{2}\right)dx_{1}\frac{dx_{2}}{2}=0,$$
(17)$$m_{11}dx_{2}-\left(m_{11}+m_{11,1}dx_{1}\right)dx_{2}+m_{21}dx_{1}-\left(m_{21}+m_{21,2}dx_{2}\right)dx_{1}+q_{1}dx_{2}\frac{dx_{1}}{2}+\left(q_{1}+q_{1,1}dx_{1}\right)dx_{2}\frac{dx_{1}}{2}=0.$$

Neglecting terms of higher order ($dx_{1}dx_{2}dx_{2},\;dx_{1}dx_{1}dx_{2}$) for small deflections, one obtains: (18)$$q_{1,1}+q_{2,2}+q=0,$$
(19)$$q_{2}=m_{12,1}+m_{22,2},$$
(20)$$q_{1}=m_{11,1}+m_{21,2}.$$

q results from the second derivatives of both the bending and torsion moments [27]:(21)$$m_{11,11}+2m_{12,12}+m_{22,22}=-q.$$

#### 3.2.1. Law of Matter

As assumed in Section 3.1, the material of the plate behaves elastically, i.e., Hooke‘s law applies (Table 2).

#### 3.2.2. Plate equations

The deflections are determined by the strains and both the forces and moments by the stresses (Section 3.2). Both are connected by the Hooke’s law (Table 2). This allows the calculation of the plate deflection as function of the acting forces and moments. To calculate the normal stress, Equations (10)–(13) are inserted into Equations (22b), (23b) and (24b):(25)$$\sigma_{11}\left(x_{1},x_{2},x_{3}\right)=-\frac{Ex_{3}}{1-\nu^{2}}\left(w_{,11}+\nu w_{,22}\right),$$
(26)$$\sigma_{22}\left(x_{1},x_{2},x_{3}\right)=-\frac{Ex_{3}}{1-\nu^{2}}\left(w_{,22}+\nu w_{,11}\right),$$
(27)$$\sigma_{12}\left(x_{1},x_{2},x_{3}\right)=-\frac{Ex_{3}}{\left(1+\nu^{\;}\right)}w_{,12},$$

The normal stresses depend linearly on the plate thickness *h*. By inserting Equations (25–27) into Equations (14a), (14b) and (14c), one achieves for the moments: (28)$$m_{11}\left(x_{1},x_{2}\right)=-\frac{Eh^{3}}{12\left(1-\nu^{2}\right)}\left(w_{,11}+\nu w_{,22}\right)=-K\left(w_{,11}+\nu w_{,22}\right),$$
(29)$$m_{22}\left(x_{1},x_{2}\right)=-\frac{Eh^{3}}{12\left(1-\nu^{2}\right)}\left(w_{,22}+\nu w_{,11}\right)=-K\left(w_{,22}+\nu w_{,11}\right),$$
(30)$$m_{12}\left(x_{1},x_{2}\right)=-\left(1-\nu \right)\frac{Eh^{3}}{12\left(1-\nu^{2}\right)}w_{,12}=-K\left(1-\nu \right)w_{,12}.$$

For simplification, the bending stiffness $K=\frac{Eh^{3}}{12\left(1-\nu^{2}\right)}$ is introduced. The shear forces have to be calculated from the equilibrium conditions from Equations (17) and (18):(31)$$q_{1}\left(x_{1},x_{2}\right)=m_{11,1}+m_{21,2}=-\frac{d}{dx_{1}}\left[K\left(w_{,11}+\nu w_{,22}\right)\right]-\frac{d}{dx_{2}}\left[K\left(1-\nu \right)w_{,12}\right],$$
(32)$$q_{2}\left(x_{1},x_{2}\right)=m_{22,2}+m_{12,2}=-\frac{d}{dx_{2}}\left[K\left(w_{,22}+\nu w_{,11}\right)\right]-\frac{d}{dx_{1}}\left[K\left(1-\nu \right)w_{,12}\right].$$ with this, the curvatures for each point of the bending surface can be expressed by the moments:(33)$$\kappa_{11}=(m_{11}-\nu m_{22})/\tilde{K},\;\kappa_{22}=(m_{22}-\nu m_{11})/\tilde{K},\;\kappa_{12}=\left(1\nu \right)m_{12}/\tilde{K},$$
with (34)$$\tilde{K}=\left(1-\nu^{2}\right)K=Eh^{3}/12.$$

By inserting the relations from Equations (28)–(30) into Equation (19), the general plate equation for a plate with small deflections can be derived:(35)$$\frac{d^{2}}{dx_{1}^{2}}\left[K\left(w_{,11}+\nu w_{,22}\right)\right]+2\frac{d^{2}}{dx_{1}dx_{2}}\left[K\left(1-\nu \right)w_{,12}\right]+\frac{d^{2}}{dx_{2}^{2}}\left[K\left(w_{,22}+\nu w_{,11}\right)\right]=q.$$
with $\Delta =\frac{d^{2}}{dx_{1}^{2}}+\frac{d^{2}}{dx_{2}^{2}}$, Equation (35) simplifies to:(36)$$\Delta \Delta w\left(x_{1},x_{2}\right)=\frac{q\left(x_{1},x_{2}\right)}{K}.$$

The moments and transverse forces are:(37)$$m_{11}\left(x_{1},x_{2}\right)=-K\left(w_{,11}\left(x_{1},x_{2}\right)+\nu w_{,22}\left(x_{1},x_{2}\right)\right),$$
(38)$$m_{22}\left(x_{1},x_{2}\right)=-K\left(w_{,22}\left(x_{1},x_{2}\right)+\nu w_{,11}\left(x_{1},x_{2}\right)\right),$$
(39)$$m_{12}\left(x_{1},x_{2}\right)=-K\left(1-\nu \right)w_{,12}\left(x_{1},x_{2}\right),$$
(40)$$q_{1}\left(x_{1},x_{2}\right)=-K\frac{d}{dx_{1}}\left(\Delta w\right),$$
(41)$$q_{2}\left(x_{1},x_{2}\right)=-K\frac{d}{dx_{2}}\left(\Delta w\right).$$

The deflection w of a plate can now be calculated by integrating Equation (36) four times, whereby the integration constants result from the specifically given boundary conditions [27].

### 3.3. Deflection of a Bending Plate due to an Out-of-Plane Force

To calculate the deflection of the bending plate in dependence of an out-of-plane force, i.e., a force acting perpendicular to the plate surface, the general plate Equation (28) must be solved with respect to the relevant boundary conditions (Figure 5). In order to calculate the forces acting on a defined area of the plate, analytical problems can be solved using a simple Fourier integral. For the development of such functions $f\left(x_{1}\right)$ and $f\left(x_{2}\right)$, they should be periodic corresponding to f(x)=f(x±nλ), with λ the period and m,n = 1.

From Reference [27] (pp. 437–442), a sinus approach with the periodical length k=a in $x_{1}$-direction and λ=b in $x_{2}$-direction is used to calculate the transverse force, where $q\left(x_{1},x_{2}\right)$ is used. m and n again correspond to the number of periods (m,n = 1). When the force p(x,y) is acting, a shear force, $q\left(x_{1},x_{2}\right)$, occurs:(42)$$\begin{array}{l} q{\left(x_{1},x_{2}\right)}_{\;}=\frac{4p_{\;}}{ab}\int_{u-\varsigma }^{u+\varsigma }\int_{v-\sigma }^{v+\sigma }\;\sin\left(m\pi \frac{k}{a}\right)\sin\left(n\pi \frac{\lambda }{b}\right)dkd\lambda \\ =\frac{p}{mn\pi^{2}}{\left[\cos\left(m\pi \frac{\kappa }{a}\right)\right]}_{u-\varsigma }^{u+\varsigma }{\left[\cos\left(n\pi \frac{\lambda }{b}\right)\right]}_{v-\sigma }^{v+\sigma }\\ =\frac{4p_{\;}}{mn\pi^{2}}\left[\cos\left(m\pi \frac{u+\varsigma }{a}\right)-\cos\left(m\pi \frac{u-\varsigma }{a}\right)\right]\left[\cos\left(n\pi \frac{v+\sigma }{b}\right)-\cos\left(n\pi \frac{v+\sigma }{b}\right)\right]. \end{array}$$Using
(43)cos(α±β)=[cos(α)cos(β)]∓[cos(α)cos(β)],
(44)sin(α±β)=[sin(α)cos(β)]±[cos(α)sin(β)],
the shear force becomes
(45)$$q{\left(x_{1},x_{2}\right)}_{\;}=\frac{16p}{\pi^{2}mn}\sin\left(m\pi \frac{u}{a}\right)\sin\left(m\pi \frac{\varsigma }{a}\right)\sin\left(n\pi \frac{v}{b}\right)\sin\left(n\pi \frac{\sigma }{b}\right).$$

By inserting m,n =1 into Equation (45), $q{\left(x_{1},x_{2}\right)}_{\;}$ corresponds to one period of the force acting onto the plate. For the analytical solution of the plate equation under load, a sinus approach is used with the period length, which is periodical length k=a in $x_{1}$-direction and λ=b in $x_{2}$-direction. m and n correspond to the number of periods in this case m,n = 1. To calculate the plate deflection $w\left(x_{1},x_{2}\right)$ from $q\left(x_{1},x_{2}\right){,}_{\;}$ the boundary conditions for an all-side-clamped plate must be taken into account:(46)$$w\left(0,0\right)=0\;w\left(a/2,b/2\right)\;\neq 0\;w^{\prime }\left(0,0\right)=0\;w^{\prime }\left(a/2,b/2\right)=0.$$

As shown in Reference [28] (pp. 91–95), the solution of the plate equation is only governed by the particulate solution. For that, a trigonometric solution approach was used to solve the plate Equation (36): (47)$$w\left(x_{1},x_{2}\right)=\frac{a^{2}b^{2}q\left(x_{1},x_{2}\right)}{K}\sin\left(n\frac{\pi }{a}x\right)\sin\left(m\frac{\pi }{b}y\right).$$

The deflection of the center of the bending plate under load is:(48)$$w\left(a/2,b/2\right)=\frac{16}{K}\frac{a^{2}b^{2}p}{mn\pi^{2}}\sin\left(m\pi \frac{u}{a}\right)\sin\left(m\pi \frac{\varsigma }{a}\right)\sin\left(n\pi \frac{v}{b}\right)\sin\left(n\pi \frac{\sigma }{b}\right)$$
and depends on the material parameters, the geometrical dimensions of the plate, and the force p($x_{1},x_{2}$). The moments $m_{xx},\;m_{xy},\;m_{yy}$ result from double derivation:(49)$$m_{xx}=\frac{d^{2}w}{dx^{2}}=\frac{1+\nu }{K}p\left(x_{1},x_{2}\right)a^{2}\sin\left(m\pi \frac{x}{a}\right)\sin\left(n\pi \frac{y}{a}\right),$$
(50)$$m_{yy}=\frac{d^{2}w}{dy^{2}}=\frac{1+\nu }{K}p\left(x_{1},x_{2}\right)a^{2}\sin\left(m\pi \frac{x}{a}\right)\sin\left(n\pi \frac{y}{a}\right),$$
(51)$$m_{xy}=\frac{d^{2}w}{\mathrm{dxd}y}=-\frac{1-\nu }{K}p\left(x_{1},x_{2}\right)a^{2}\cos\left(m\pi \frac{x}{a}\right)\cos\left(n\pi \frac{y}{a}\right).$$

For a square plate shape with a=b, it follows [29,30]
(52)$$\frac{d^{2}w}{dx^{2}}=\frac{d^{2}w}{dy^{2}}=m_{xx}=m_{yy}.$$

### 3.4. Deflection of a Bending Plate due to the Bimorph Effect

To calculate an approximate solution for the deflection of a bending plate under the influence of an in-plane force, some considerations have to be made: As shown in Reference [31], the effect of a bimorph bending element can be considered as a conventional monomorph bending element with an additionally acting internal moment source (affecting the bending behavior), as well as an internally generated in-plane force, due to the effective thermal length expansion.This in-plane force is comparable to the force $q_{1}\left(x_{1},x_{2}\right)$ and $q_{2}\left(x_{1},x_{2}\right)$ from Equations (14e)–(14f) for the case of the plate from Figure 5. For that reason, the resulting deflection of a thin rectangular plate with a bimorph element in the center should follow the same function as of Equation (47).The latter applies only when the mentioned internally generated in-plane force is used. Therefore, this force has to be considered afterwards in a next step.In our application, the bimorph comprises the rectangular silicon plate and the humidity-sensitive hydrogel in the middle. Because the Young’s modulus of hydrogel is several orders of magnitude lower than that of silicon, its influence on both the bending and the axial stiffness is neglectable. Therefore, the bimorph bending plate can still be considered to be homogeneous and with a constant thickness.

As illustrated in Figure 2d, the deflection of the bimorph bending plate can now be calculated in dependence on the transverse forces $q(x_{1}$) and $q(x_{2}$). Due to the plate symmetry, it can be assumed that $q(x_{1}$) = $q(x_{2}$).

As given in Figure 6, the (silicon) plate has a thickness of $h_{1}$ and a Young´s modulus $E_{1}$, whereas the patterned (hydrogel) layer is characterized by $h_{2}$ and $E_{2}$. According to Reference [31], the neutral axis is located at
(53)$$x_{s}=\frac{1}{2}\frac{E_{1}h_{1}^{2}-E_{2}h_{2}^{2}}{E_{1}h_{1}^{\;}+E_{2}h_{2}^{\;}}.$$

The integral force $F_{1}$ in $x_{1}$-direction results approximately from: (54)$$F_{1}=-l_{B}\int_{x_{s}-h_{2}}^{x_{s}+h_{2}}\sigma_{13}dx_{3}=-l_{B}(\alpha_{1}E_{1}h_{1}^{\;}+\alpha_{2}E_{2}h_{2}^{\;})\Lambda .$$

Here, $\alpha_{1}$ and $\alpha_{2}$ are the expansion coefficients from the plate and the patterned layer with respect to a length expansion quantity Λ (e.g., swelling due to humidity or temperature).

The transverse force $F_{1}$ depends on the bimorph length $l_{B}=\varsigma$, the thickness $h{,}_{\;}$ the Young’s modulus E, the expansion coefficient α, and the influence quantity Λ. The indices 1 and 2 are assigned to the respective material (Figure 6). The material with index 1 (silicon) does not undergo any elongation due to humidity (α_1_ = 0) by Λ, thus simplifying Equation (55) to:(55)$$F_{1}=l_{B}\left(\alpha_{2}E_{2}h_{2}^{\;}\right)\Lambda .$$

The length-related shear force of a plate is calculated according to Equation (14e):(56)$$q_{1}=\frac{F_{1}}{l}=\frac{l_{B}}{l}\left(\alpha_{2}E_{2}h_{2}^{\;}\right)\Lambda ,$$
with the total length $a=l_{\;}$ = $l_{B}+l_{P}$ (Figure 6). The shear force thus depends exclusively on the properties of the patterned film and the coverage ratio $l_{B}/l$. According to the derivation of the deflection of the plate in Section 3.4, Equations (42)–(48), it now yields for the bimorph plate:(57)$$q{\left(x_{1},x_{2}\right)}_{\;}=4p\int_{u-\varsigma }^{u+\varsigma }\int_{v-\sigma }^{v+\sigma }\;\sin\left(m\pi \frac{k}{a}\right)\sin\left(n\pi \frac{\lambda }{b}\right)dkd\lambda .$$

The deflection in the center is given for $x_{1}=a/2$ and $x_{1}=b/2$: (58)$$w\left(a/2,b/2\right)=\frac{16}{K}\frac{a^{2}b^{2}p}{mn\pi^{2}}\sin\left(m\pi \frac{u}{a}\right)\sin\left(m\pi \frac{\varsigma }{a}\right)\sin\left(n\pi \frac{v}{b}\right)\sin\left(n\pi \frac{\sigma }{b}\right).$$

To calculate the deflection of the bending plate depending of the ratio $l_{B}/l$ (the coating degree), the shear force must be adapted to the plate. The length $l_{B}$ is variable. Since the structure is symmetrical, the shear force $q_{2}\;$ parallel to the $x_{2}$-axis is equal to $q_{1}$ parallel to the $x_{1}$-axis:(59)$$q_{1}=q_{2}=\frac{l_{B}}{l}\left(\alpha_{2}E_{2}h_{2}^{\;}\right)\Lambda .$$

Since the shear force affects a part of the plate surface, the force is calculated by interpolation in $x_{2}$-direction. Using the integral limits from Figure 4, the shear force in $x_{1}$- and $x_{2}$-direction for a plate is:(60)$$q{\left(x_{1},x_{2}\right)}_{\;}=\frac{2l_{B}\left(\alpha_{2}E_{2}h_{2}^{\;}\right)\Lambda_{\;}}{ab}\int_{u-\varsigma }^{u+\varsigma }\int_{v-\sigma }^{v+\sigma }\;\sin\left(m\pi \frac{\kappa }{a}\right)\sin\left(n\pi \frac{\lambda }{b}\right)d\kappa d\lambda \;=\frac{2l_{B}\left(\alpha_{2}E_{2}h_{2}^{\;}\right)\Lambda }{mn\pi^{2}}{\left[\cos\left(m\pi \frac{\kappa }{a}\right)\right]}_{u-\varsigma }^{u+\varsigma }{\left[\cos\left(n\pi \frac{\lambda }{b}\right)\right]}_{v-\sigma }^{v+\sigma },$$
(61)$$w\left(x_{1},x_{2}\right)=\frac{16}{K}\frac{l_{B}a^{4}\left(\alpha_{2}E_{2}h_{2}^{\;}\right)\Lambda }{mn\pi^{2}}\sin\left(m\pi \frac{u}{a}\right)\sin\left(m\pi \frac{\varsigma }{a}\right)\sin\left(n\pi \frac{v}{b}\right)\sin\left(n\pi \frac{\sigma }{b}\right)\sin\left(m\pi \frac{x}{a}\right)\sin\left(n\pi \frac{y}{a}\right).$$

For the calculation of the deflection at the center of the plate, Equation (47) is used with $x_{1}\;$= a/2, $x_{2}$ = b/2 and with Equation (61), as follows:(62)$$w\left(a/2,b/2\right)=\frac{16}{K}\frac{a^{4}l_{B}\left(\alpha_{2}E_{2}h_{2}^{\;}\right)\Lambda }{mn\pi^{2}}\sin\left(m\pi \frac{u}{a}\right)\sin\left(m\pi \frac{\varsigma }{a}\right)\sin\left(n\pi \frac{v}{b}\right)\sin\left(n\pi \frac{\sigma }{b}\right).$$

The moments follow by double derivation:(63)$$m_{\mathrm{xx}}=\frac{d^{2}w}{dx^{2}}=\frac{1+\nu }{K}q\left(x_{1},x_{2}\right)a^{2}\sin\left(m\pi \frac{x}{a}\right)\sin\left(n\pi \frac{y}{a}\right),$$
(64)$$m_{\mathrm{yy}}=\frac{d^{2}w}{dy^{2}}=\frac{1+\nu }{K}q\left(x_{1},x_{2}\right)a^{2}\sin\left(m\pi \frac{x}{a}\right)\sin\left(n\pi \frac{y}{a}\right),$$
(65)$$m_{\mathrm{xy}}=\frac{d^{2}w}{dxdy}=-\frac{1-\nu }{K}q\left(x_{1},x_{2}\right)a^{2}\cos\left(m\pi \frac{x}{a}\right)\cos\left(n\pi \frac{y}{a}\right).$$

If, instead of the center of the plate, the outer part of the rectangular plate is covered (Figure 7), the corresponding integration limits in Equation (60) have to be changed: (66)$$q{\left(x_{1},x_{2}\right)}_{\;}=\frac{2\cdot l_{B}\left(\alpha_{2}E_{2}h_{2}^{\;}\right)\Lambda_{\;}}{ab}\int_{u-\varsigma }^{u+\varsigma }\int_{v-\sigma }^{v+\sigma }\;\sin\left(m\pi \frac{k}{a}\right)\sin\left(n\pi \frac{\lambda }{b}\right)dkd\lambda ,$$
and the plate equation results in:(67)$$w\left(x_{1},x_{2}\right)=\frac{16}{K}\frac{a^{4}l_{B}\left(\alpha_{2}E_{2}h_{2}^{\;}\right)\Lambda }{\pi^{2}}\sin\left(\pi \frac{u}{a}\right)\sin\left(\pi \frac{\varsigma }{a}\right)\sin\left(\pi \frac{v}{b}\right)\sin\left(\pi \frac{\sigma }{b}\right)\sin\left(\pi \frac{x_{1}}{a}\right)\sin\left(\pi \frac{x_{2}}{b}\right).$$

By inserting $x_{1}=a/2$ and $x_{2}=b/2$ into Equation (67), the deflection of the center of the plate results in:(68)$$w\left(a/2,b/2\right)=\frac{16}{K}\frac{a^{4}l_{B}\left(\alpha_{2}E_{2}h_{2}^{\;}\right)\Lambda }{\pi^{2}}\sin\left(m\pi \frac{u}{a}\right)\sin\left(m\pi \frac{\varsigma }{a}\right)\sin\left(n\pi \frac{v}{b}\right)\sin\left(n\pi \frac{\sigma }{b}\right)\cdot 1\cdot 1.$$

Again, the strong influence of the coverage ratio ς/a and $l_{B}/l$, respectively, can be seen.

Figure 8 compares this influence for the both cases (Figure 5 and Figure 7), where a silicon plate is covered by a patterned hydrogel layer in the middle and at the outer part at the clamping of the plate. For reasons of simplification, a square shape was considered (a=b, ς=σ). The calculations are based on Equations (48) and (62), and use the material and geometrical parameters from Table 3. Figure 8 shows that a maximum deflection occurs for a coverage ratio ς/(a/2) of ca. 0.55. This confirms our results from Reference [4] for a simplified lumped model of a bimorph bending plate. 

### 3.5. Deflection of a Plate with Axial In-Plae Compresion Force

As mentioned in Section 2, sensor switches as considered in this work should show an S-shaped hysteresis. This can be achieved—in conjunction with a corresponding pre-deflection of the bending plate—by applying an axial compressive force (Figure 9). For instance, the deposition of a silicon oxide layer on the top surface at the silicon bending plate could create such a compressive stress, since SiO_2_ is deposited at temperatures much above room temperature (RT). Due to the smaller coefficient of thermal expansion of SiO_2_ in comparison with Si (α_SiO2_ = 5.6 × 10^−7^ K^−1^ < α_Si_ = 2.3 × 10^−6^ K^−1^), cooling down to RT leads to this targeted compressive force [23,29]. The axial compression then leads to a pre-deflection of the bending plate (Figure 9). 

From the boundary conditions, the pre-deflection of a plate can be expressed by a bending plate that is all-side-axially clamped and axially compressed by a length-related force *n*. This leads to a buckling of the plate. For this case, the plate equation for an axially compressed plate shall be developed. All assumptions from Section 3.2 regarding the plate are still valid. Only the equilibrium conditions from Equation (13) for the transverse line forces $n_{11}$ and $n_{22}$, which act perpendicular to the plate cross-section area, are now different from zero [27]:(69a)$$n_{11}=\int_{-\frac{h}{2}}^{\frac{h}{2}}\sigma_{11}dz\neq 0,$$
(69b)$$n_{22}=\int_{-\frac{h}{2}}^{\frac{h}{2}}\sigma_{22}dz\neq 0,$$
(69c)$$n_{33}=\int_{-\frac{h}{2}}^{\frac{h}{2}}\sigma_{33}dz=0.$$

The plate Equation (36) then yields:(70)$$K\Delta \Delta w+n_{11}\frac{d^{2}w}{dx_{1}^{2}}+n_{22}\frac{d^{2}w}{dx_{2}^{2}}=0.$$

The line forces $n_{11}$ and $n_{22}$ arise from the normal stresses $\sigma_{11}$ and $\sigma_{22}$, corresponding to the two axes of the coordinate system. 

The general solution approach for Equation (70) is:(71)$$w=C\cdot \sin\left(n\frac{\pi }{a}x_{1}\right)\sin\left(m\frac{\pi }{b}x_{2}\right),$$And, with α=nπ/a  and β=mπ/b, it follows:(72)$$w=C\cdot \sin\left(\alpha x_{1}\right)\sin\left(\beta x_{2}\right).$$

By inserting Equation (72) into Equation (70), it follows:(73)$$\left[K{\left(\alpha^{2}+\beta^{2}\right)}^{2}-n_{11}\alpha^{2}-n_{22}\beta^{2}\right]\cdot \sin\left(\alpha x_{1}\right)\sin\left(\beta x_{2}\right)=0,$$
and, after rewriting:(74a)$$n_{11}=K\frac{{\left(\alpha^{2}+\beta^{2}\right)}^{2}}{\alpha^{2}+\beta^{2}}.$$

A symmetrically clamped plate and homogeneous axial loads lead to the ratio $n_{22}$/$n_{11}=1$. The critical force that leads to a pre-deflection of the bending plate depends from the edge lengths of the plate.

The still valid boundary conditions from Equation (4) are also fulfilled for this solution. Thus, with α = β = 1, i.e., for a square plate, it yields:(74b)$$n_{11}=\frac{2\pi^{2}K}{a^{2}}.$$

Here, the transverse force $n_{11}$ corresponds to the axial force from Figure 9. 

Figure 10 shows the pre-deflection of a plate as a function of the axial force. For n/K<1, no deflection occurs, whereas, for n/K>1, a deflection can be seen [29]. This corresponds to the buckling load well-known from buckling bars. For small pre-deflections of a bending plate, the resulting pre-deflection can be approximately described via a linear term $\varphi_{0}$ of the pre-deflection angle (Figure 10):(75)$$\frac{n}{K}\approx \varphi_{0}$$

### 3.6. Deflection of a Bimorph Bending Plate with Axial In-Plane Compression Force

Taking into account the assumptions for $q\left(x_{1},x_{2}\right)$ from Equation (74), $n_{11}$ and $n_{22}$ are added to Equation (36) to calculate the deflection w of a bimorph plate with both axial load and in-plane force. Thus, the plate equation is rewritten to:(76)$$K\Delta \Delta w+\Delta w=q{\left(x_{1},x_{2}\right)}_{\;}$$
and is equal to
(77)$$K\Delta \Delta w\left(x_{1},x_{2}\right)+n_{11}\frac{d^{2}w}{dx_{1}^{2}}+n_{22}\frac{d^{2}w}{dx_{2}^{2}}=q\left(x_{1},x_{2}\right).$$
with the general solution, (78)$$w=C\cdot \sin\left(n\frac{\pi }{a}x\right)\sin\left(m\frac{\pi }{b}y\right),$$It follows, according to Equation (67):(79)$$w\left(a/2,b/2\right)=\frac{16}{K}\frac{a^{4}l_{B}E_{2}h_{2}^{\;}\alpha_{2}\Lambda }{mn\pi^{2}\left(1-n/K\right)}\sin\left(m\pi \frac{u}{a}\right)\sin\left(m\pi \frac{\varsigma }{a}\right)\sin\left(n\pi \frac{v}{b}\right)\sin\left(n\pi \frac{\sigma }{b}\right).$$

The solution of this plate equation includes the ratio *n/K* with n the normal force perpendicular to the plate cross-section and the plate stiffness *K* [30]. To illustrate this ratio, the denominator of Equation (78) is extended by (2π^2^−*n/K*) and leads to:(80)$$w\left(a/2,b/2\right)=\frac{16}{K}\frac{a^{4}l_{B}E_{2}h_{2}^{\;}\alpha_{2}\Lambda \cdot \left(1-n/K\right)}{mn\pi^{2}{\left(1-\frac{n}{K}\right)}^{2}}\sin\left(m\pi \frac{u}{a}\right)\sin\left(m\pi \frac{\varsigma }{a}\right)\sin\left(n\pi \frac{v}{b}\right)\sin\left(n\pi \frac{\sigma }{b}\right).$$

The strain $\varepsilon =\alpha_{2}\Lambda$ of the hydrogel is caused by the relative humidity R.H.=Λ, $\alpha_{2}$ denotes the corresponding length extension coefficient due to swelling. ε was measured via a tensile testing machine [7]. The hydrogel swelling provides the mechanical energy to deflect the bending plate by the deflection w. w itself is directly proportional to the deflection angle φ. Therefore, φ in Figure 11 is directly related to w and, hence, to the relative humidity and the related strain, respectively.

For small deflections of the bending plate, the deflection angle is linearly proportional to φ, where for larger deflections a cubical relationship $\varphi^{3}$ can be seen:(81)$$\varepsilon =\left(\alpha_{2}\Lambda \right)\approx \left(0.045\Lambda +\left(\frac{1}{6}\right){\left(0.045\Lambda \right)}^{3}\right)\approx \left(\varphi +\left(\frac{1}{6}\right)\varphi^{3}\right).$$

This again confirms the nonlinear relationship derived in Reference [8] for a bending beam. Inserting Equation (81) into Equation (80) leads to the deflection of the center of the square bending plate (a/2 = b/2):(82)$$w\left(a/2,b/2\right)=\frac{16}{K}\frac{a^{4}l_{B}\left(E_{2}h_{2}^{\;}\right)\left(\varphi +\varphi^{3}\right)\cdot \left(1-\varphi_{0}\right)}{mn\pi^{2}{\left(1-\frac{n}{K}\right)}^{2}}=\frac{16}{K}\frac{a^{4}l_{B}\left(E_{2}h_{2}^{\;}\right)(\varphi +\left(\frac{1}{6}\right)\varphi^{3}-\varphi \varphi_{0}-\varphi^{3}\varphi_{0})}{mn\pi^{2}{\left(1-\frac{n}{K}\right)}^{2}}.$$

Figure 12 shows the calculated curve progression in dependence of the pre-deflection *n/K* and $\varphi_{0}$. For increasing values of $\varphi_{0}$, the width of the S-shaped curve increases steadily. For $\varphi_{0}$ < 1, no S-shaped curve progression can be recognized, and, for $\varphi_{0\;}$> 1, a deflection with an S-curve progression can be recognized. 

## 4. Fabrication of a Si-Based Sensor Switch with Switching Hysteresis

As described in Section 2 and Section 3.6 the targeted switching hysteresis requires both a pre-deflection and an axial (in-plane) compressive stress. This can be achieved by a second bimorph element in the center of the plate that leads—in the switched-off state—to the required open contact. 

### 4.1. Set-Up

In the case of silicon plates, both the pre-deflection and the axial pre-stress can be achieved by depositing an SiO_2_ layer in the center of the plate. SiO_2_ has a coefficient of thermal expansion (5.6 × 10^−7^ K^−1^) that is much smaller than that one of Si (2.3 × 10^−6^ K^−1^) [31]. After cooling down the structure from higher temperatures to room temperature, both the targeted pre-deflection and the compressive stress in the plane are established (Figure 13). 

The SiO_2_ layer was deposited on the silicon bending plate using a PE-CVD (plasma-enhanced chemical vapor deposition) process (Plasmalab80Plus, Oxford Plasma Technology, Yatton, UK). The deposition temperature amounted to 300 °C, the process time was 4 h, and the resulting layer thickness was 3.5 µm. Since the optimum deposition ratio should be ca. 0.5 of the plate size (Figure 8), the SiO_2_ size was chosen as 2 × 2 mm^2^ for the Si plate. The resulting pre-deflection of the silicon plate (20 µm thickness, size 3.75 × 3.75 mm²) was determined as 20 µm (Figure 13) [24].

### 4.2. Pattering of the Hydrogel Layer 

For structuring the hydrogel layer, a silicone stamp is used. It was punched out of a Sylgard-184 (Sigma-Aldrich, St. Louis, MO, USA) hydrogel layer by using a biopuncher (Harris Uni-Core, Carl Roth, Karlsruhe, Germany) with a diameter of 1.5–3.5 mm. The silicone elastomer kit of Sylgard-184 consists of two liquid components, the base Polydimethylsiloxan (PDMS) and the cross-linking agent. Both components were mixed in a 10:1 ratio by weight. 

To ensure adhesion between the hydrogel and the bending plate, a thin adhesion layer was coated on the cleaned silicon plate using a 3% 3-aminopropyltriethoxysilane solution (C9H23NO3Si, APTES) (Carl Roth, Karlsruhe, Germany) [32]. It was cured and activated at 90 °C for about 10 min. Then the silicone stamp was attached in the center of the silicon bending plate. Figure 14 shows the structuring process of the hydrogel layer via the stamp technique. 

For the humidity-sensitive hydrogel two polymer powders, 15 wt% poly(vinyl alcohol) (PVA, molecular mass Mw = 89,000–98,000) and 7.5 wt% poly(acryl acid) (PAA, Mw = 450,000), were used. Both materials were purchased from Sigma-Aldrich (St. Louis, MO, USA). The powders were solved in deionized water and mixed for several hours at 80 °C until a homogenous solutions was formed. Afterward, both polymer solutions were mixed in a mass ratio of 4:1 (PVA:PAA) [7,23,33]. 

The degassed polymer solution was filled with a pipette into the resulting cavity with a height of about 370 µm. To produce the hydrogel layer, the cavity is filled up with a polymer solution to the upper chip edge. Since the polymer solution has the same polymer concentration and only the coating area is varied via the silicone stamp diameter, the layer thickness is reproducible (Figure 14). After the solvent water was evaporated, the silicone stamp was removed. Depending on the viscosity of the polymer solution, which is mainly influenced by the water content, a layer of about 30 µm in thickness was formed. The hydrogel results from the crosslinking of the now dry polymer solution at a temperature of 130 °C for 20 min Reference [34,35,36,37,38,39]. 

## 5. Experimental

### 5.1. Measerment Set-Up

To prove the switching hysteresis of the hydrogel-covered bending plate, the deflection of the bimorph bending plate in dependence of the relative humidity was optically examined with a 3D-profilometer (µScan NanoFocus). The measurement took place in a humidity chamber where the humidity was generated by a bubble system. For humidity monitoring, a humidity sensor (HYTELOG USB, B+B sensors) was used (Figure 15). 

The humidity content was varied successively between 20% R.H. and 97% R.H. by 10% R.H. steps. To ensure the reproducibility of each measurement, the humidity was maintained for 25 min after each humidity change [4]. 

### 5.2. Results

From the mechanical model with axial compression (Equation (82)), the S-shape curve progression shows an increase in the width of the hysteresis, which depends on the axial compression. The pre-deflection was successfully achieved due to the structured SiO_2_ layer onto the Si-plate surface. From the measured results, the pre-deflected and partly with hydrogel covered Si-plate shows a snap-through switching behavior. The switching threshold values amounted to 86% R.H. and 53% R.H., respectively, for increasing and decreasing humidity (Figure 16). Above 86% R.H., the bending plate abruptly switches from the pre-deflected side to the opposite side. At 53% R.H., by changing from the humid to the dry environment, the plate switches back to the initial state, demonstrating successfully the switching hysteresis. 

To validate the mechanical model, the deflection of the center of a bending plate was calculated with Equation (82) and with $l_{B}/l$ = 0.5 and n/K = 1.5. The material and geometrical parameter were taken from Table 3 and $\varphi_{0}=0.00057{}^{\circ}$ from Figure 13. Figure 16 shows the calculated deflection. The deflection angle $\varepsilon =\left(\alpha_{2}\Lambda \right)\approx \left(\varphi +\left(\frac{1}{6}\right)\varphi^{3}\right)$ of the bending plate was taken from the measured results from Figure 16 and inserted in Equation (82):(83)$$w\left(a/2,b/2\right)=\frac{16}{K}\frac{a^{4}l_{B}\left(E_{2}h_{2}^{\;}\right)\left(\varphi +\varphi^{3}\right)\cdot \left(1-\varphi_{0}\right)}{mn\pi^{2}{\left(1-\frac{n}{K}\right)}^{2}}=\frac{8}{K}\frac{a^{4}\left(E_{2}h_{2}^{\;}\right){\left(\varphi -0.00057\varphi +\left(\frac{1}{6}\right)\varphi^{3}\right)}_{\;}}{\pi^{2}{\left(1-1.5\right)}^{2}}.$$

If the humidity reaches the threshold φ_high_, then the deflection changes suddenly from a positive value (open contact) to a negative one (closed contact). This occurs abruptly, leading to a fast switching of the contact. Vice versa, by decreasing humidity below the lower threshold φ_low_, the contact will be opened in the same sudden manner. The higher the switching hysteresis φ_high_–φ_low_ is, the less the switch is prone to oscillations due to small changes around the threshold. 

## 6. Conclusions

This work deals with the miniaturization and development of a rectangular-shaped, silicon-based sensor switch with switching hysteresis for humidity sensing. The hysteresis is needed for the safe and reliable switching, i.e., fast opening and closing of the contacts. 

Based on the plate theory, an analytical mechanical model was derived to calculate the deflection w depending on the coating degree, i.e., of the size ratio of the partly hydrogel-coated bending plate part to the total size of the bending plate. It could be shown that a maximum deflection is achieved at half cover, i.e., when the stiffness of the hydrogel-coated part and the uncoated part are approximately equal. The calculated deflection showed a deviation of ca. 10% in comparison to the measured value. The cause of such a deviation results from the assumptions and approximations, which were taken into account in particular considering small deflections. 

For the implementation of the switching hysteresis, the mechanical model was extended by an axial compression. From this, it could be shown that the width of the switching hysteresis is adjustable via geometric and technological parameters influencing the axial compression force. To prove the mechanical model, a silicon based sensor switch was manufactured, and the switching hysteresis was successfully demonstrated. Pre-deflection and hysteresis were achieved by the deposition of an SiO_2_ layer in the center of the rectangular silicon plate. In comparison to the measured results, the calculated deflection shows a deviation in the deflection amplitude of ca. 20%. 

Experiments used MEMS-based silicon plates, where both the pre-deflection and the axial pre-stress were achieved by depositing an SiO_2_ layer in the center of the plate. Because SiO_2_ has a coefficient of thermal expansion that is much smaller than that of Si and is deposited at high temperatures and cooling down to room temperature causes the needed compressive in-plane stress. The SiO_2_ layer was deposited using a PE-CVD process. The resulting pre-deflection of the 20 µm thick silicon plate with a size of 3.75 × 3.75 mm² amounted to 20 µm.

The axial compression caused the predicted S-shape curve progression with the targeted snap-through switching behavior. The switching threshold values amounted to 86% R.H. and 53% R.H., respectively, for increasing and decreasing humidity.

## Figures and Tables

**Figure 1 micromachines-11-00569-f001:**
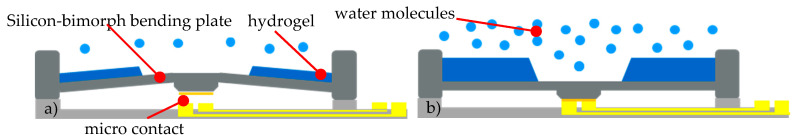
Basic operational principle of a humidity threshold sensor switch, consisting of a bimorph bending plate with patterned hydrogel layer and micro-contacts, (**a**) open and (**b**) closed (adapted from Reference [7]).

**Figure 2 micromachines-11-00569-f002:**
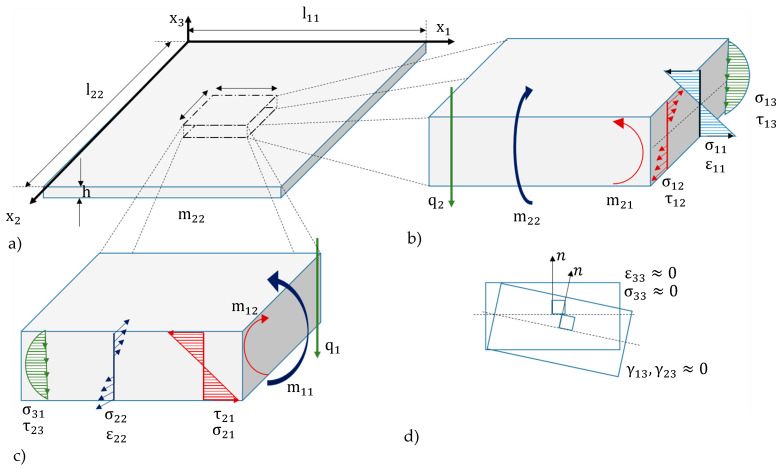
Rectangular thin plate with high bending stiffness (**a**); (**b**,**c**) Thickness-dependent stress and strain distributions and plate element with internal moments due to the stress distribution along the thickness *h*; (**d**) Undeflected and deflected plate element, (adapted from Reference [27]).

**Figure 3 micromachines-11-00569-f003:**
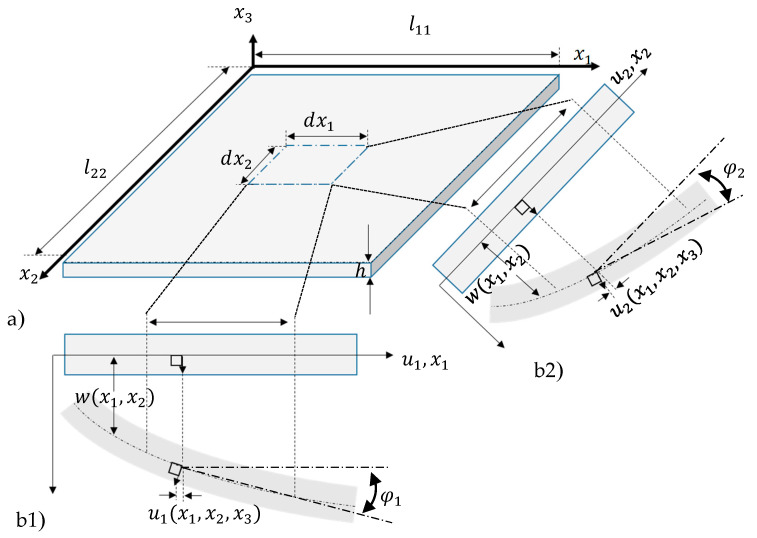
(**a**) Plate with thickness *h* and cut-out plate element, deformed and undeformed state in (**b1**) $x_{1}$- and (**b2**) $x_{2}$-direction (adapted from Reference [27]).

**Figure 4 micromachines-11-00569-f004:**
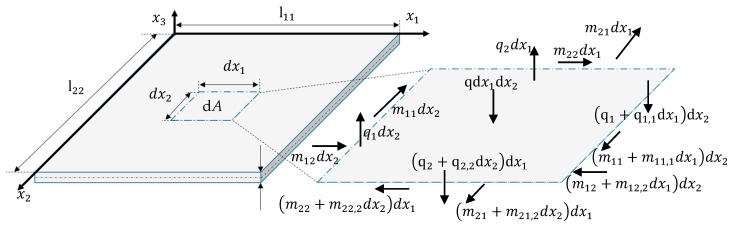
Moments and forces acting at a plate part with $dA=dx_{1}dx_{2}$ (adapted from Reference [27]).

**Figure 5 micromachines-11-00569-f005:**
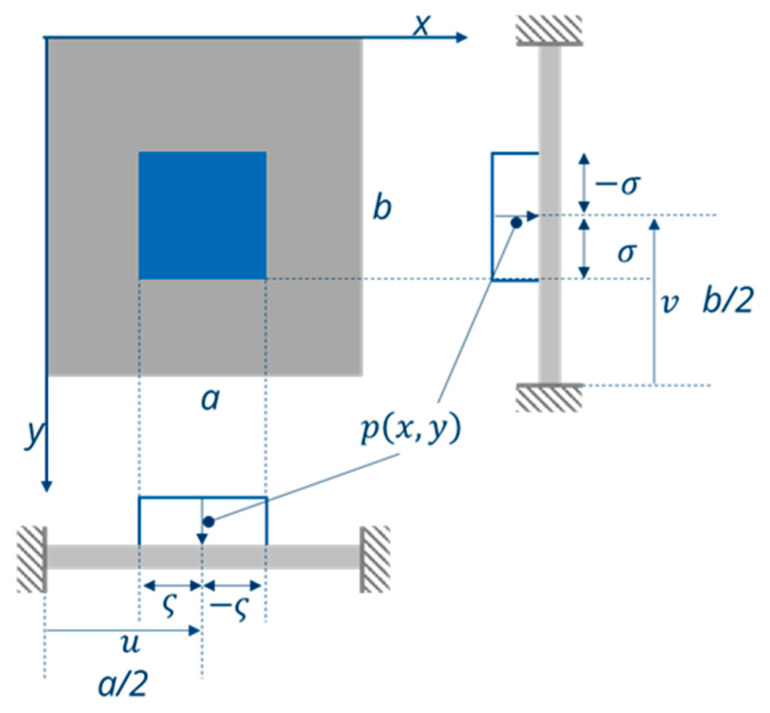
Entirely clamped rectangular plate (grey) with area-related out-of-plane force *p*(*x*,*y*) acting on the middle of the plate (blue area). The center of this area is located at *u* = *a*/2 and *v* = *b*/2, and its size is 2σ × 2ς (adapted from Reference [27]).

**Figure 6 micromachines-11-00569-f006:**
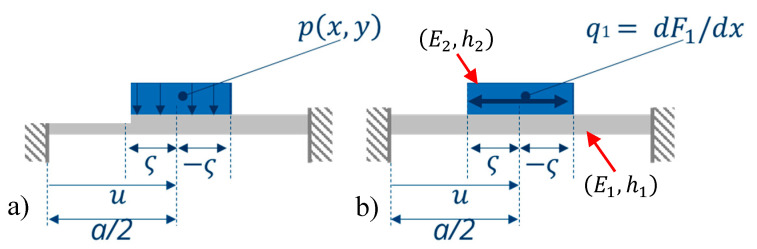
Analogy between (**a**) the rectangular plate from Figure 5 and (**b**) a bimorph plate, where the bimorph is located in the plate center.

**Figure 7 micromachines-11-00569-f007:**
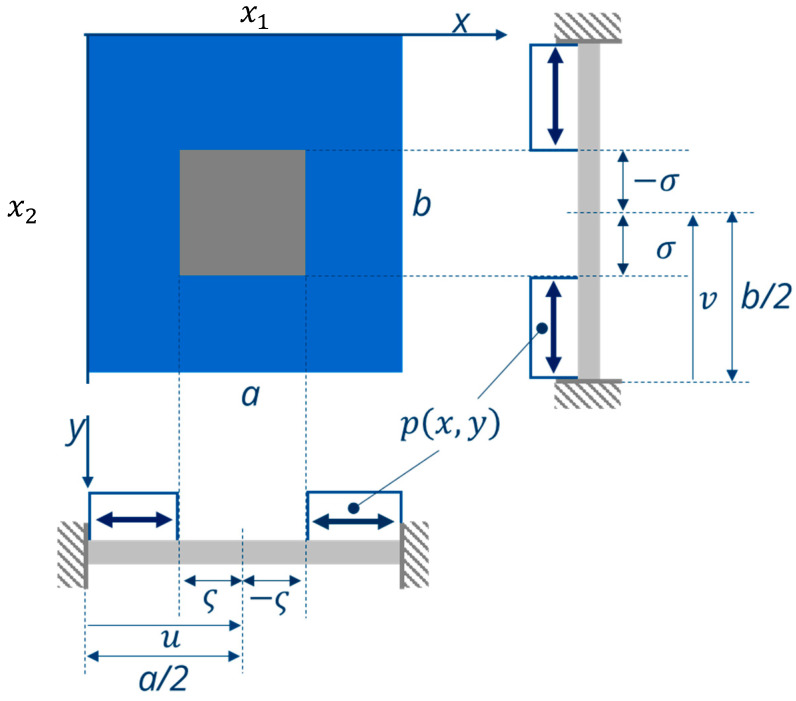
All-side-clamped rectangular bending plate (grey) with edge lengths a and b, with the outer side covered with a swellable layer.

**Figure 8 micromachines-11-00569-f008:**
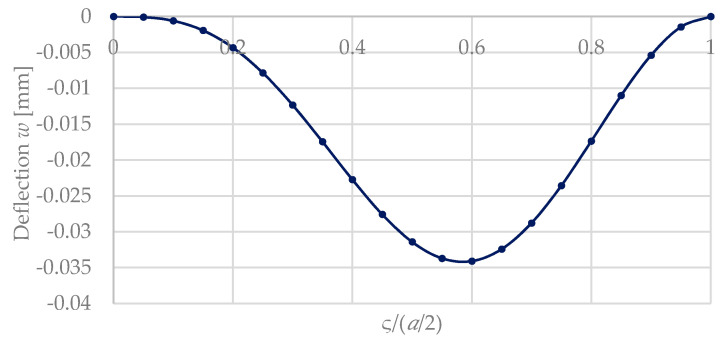
Deflection w of the plate center $w\left(\frac{a}{2},\frac{b}{2}\right)$ with a=b of a square plate calculated with Equation (68) in dependence of the coating degree (ς)/(a/2) for a symmetrically coated layer (ς = σ). Geometrical parameters as in Table 3.

**Figure 9 micromachines-11-00569-f009:**
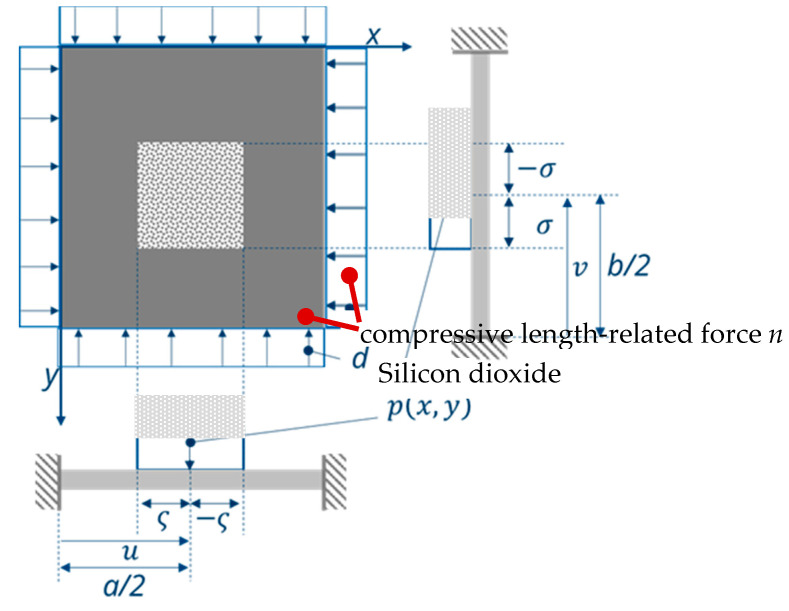
Fully clamped rectangular bending plate (grey) with structured silicon dioxide (SiO_2_) layer (light grey).

**Figure 10 micromachines-11-00569-f010:**
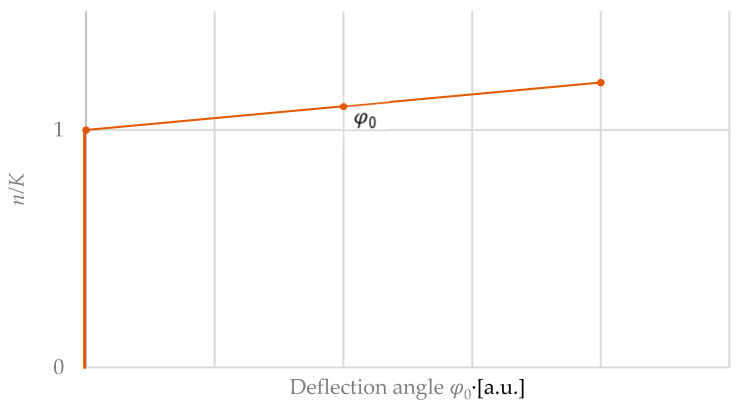
Deflection of a clamped rectangular bending plate as a function of the relative axial force *n/K* [29].

**Figure 11 micromachines-11-00569-f011:**
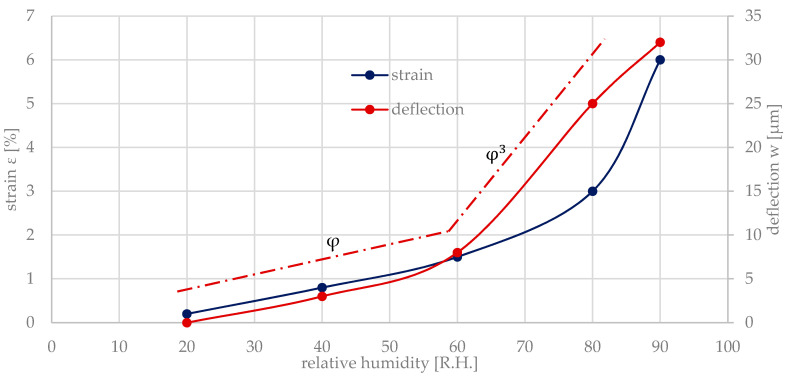
Strain ε in the hydrogel as a function of the relative humidity and the deflection *w* with a linear and cubical term of the angle $\varphi^{3}$ (adapted from Reference [7]).

**Figure 12 micromachines-11-00569-f012:**
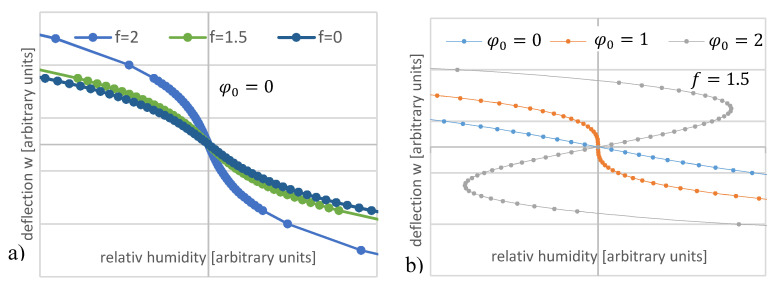
Calculated deflection w of a rectangular bending plate in dependence of the deflection angle φ from Equation (81): (**a**) with the pre-deflection angle $\varphi_{0}=0$ and variable f; (**b**) with the variable pre-deflection angle $\varphi_{0}$ and f = 1.5.

**Figure 13 micromachines-11-00569-f013:**
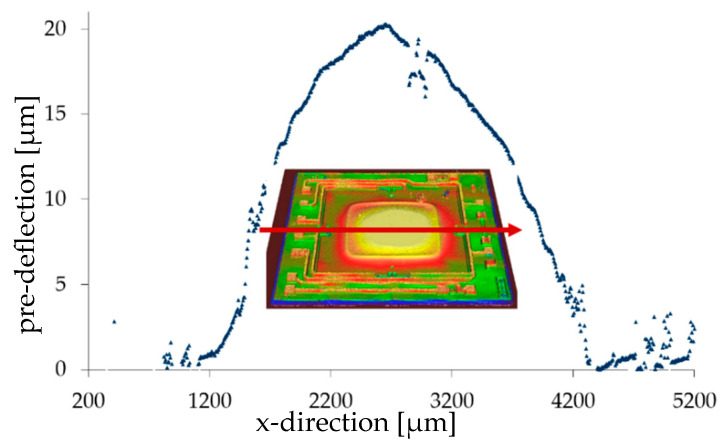
Pre-deformation of the Si bending plate partly covered with an SiO_2_ layer. The pre-deflection was measured optically with a 3D-profilometer (µScan; NanoFocus, Oberhausen, Germany) along the red line.

**Figure 14 micromachines-11-00569-f014:**
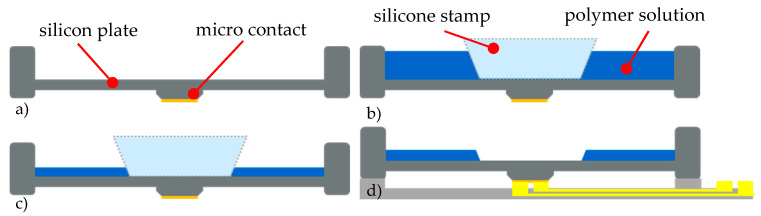
Patterning of the hydrogel layer atop the silicon bending plate by using a silicone stamp: (**a**) initial state, (**b**) silicone stamp attached to the bending plate, cavity filled with polymer solution, (**c**) after evaporation of the solvent from the polymer, (**d**) resulting hydrogel layer after the stamp was removed and the polymer was cross-linked 8 (adapted from Reference [7]).

**Figure 15 micromachines-11-00569-f015:**
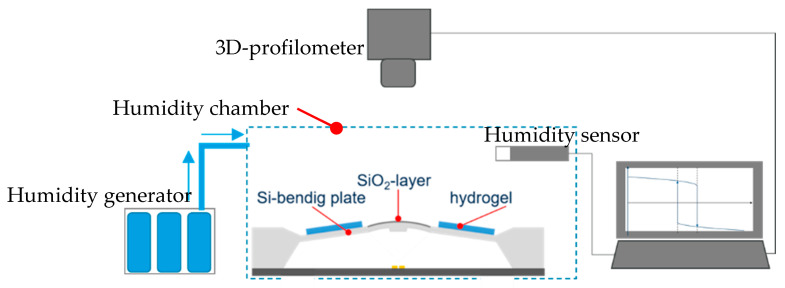
Set-up comprising the humidity chamber, the humidity sensor, and the humidity generator.

**Figure 16 micromachines-11-00569-f016:**
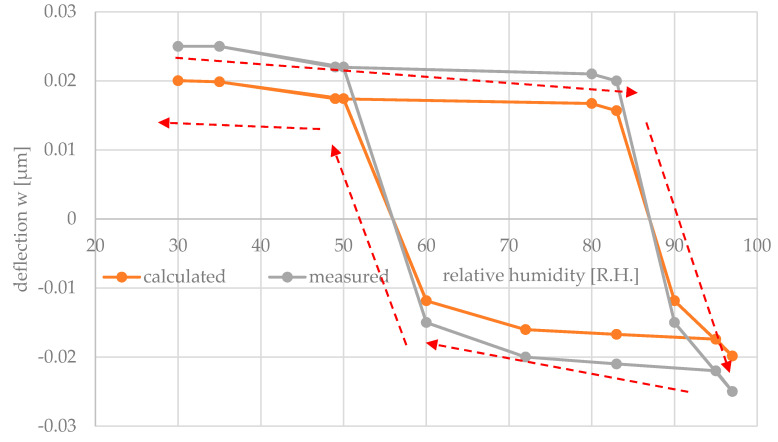
Measured and calculated deflection of a bending plate partly covered with an SiO_2_-layer and a hydrogel layer as a function of the relative humidity, with the pre-deflection angle $\varphi_{0}$ = 0.00057.

**Table 1 micromachines-11-00569-t001:** Bending moments, torsional moments, transverse forces, and the corresponding boundary conditions.

Length-Related Quantities	Equation	Unit	At Location	
Bending moments	$m_{11}\left(x_{1},x_{2}\right)=\int_{-\frac{h}{2}}^{\frac{h}{2}}\sigma_{11}x_{3}dz$	[N]	$x_{1}\;$= const.	(14a)
	$m_{22}(x_{1},x_{2}=\int_{-h/2}^{h/2}\sigma_{22}x_{3}dz$	[N]	$x_{2}\;$= const.	(14b)
Torsion moment	$m_{12}\left(x_{1},x_{2}\right)=\int_{-h/2}^{h/2}\sigma_{12}x_{3}dz$	[N]	$x_{1}\;$= const.	(14c)
	$m_{21}\left(x_{1},x_{2}\right)=\int_{-h/2}^{h/2}\sigma_{21}x_{3}dz$	[N]	$x_{2}\;$= const.	(14d)
Transverse force	$q_{1}\left(x_{1},x_{2}\right)=\int_{-h/2}^{h/2}\sigma_{13}dz$	[N/m]	$x_{1}\;$= const.	(14e)
	$q_{2}\left(x_{1},x_{2}\right)=\int_{-h/2}^{h/2}\sigma_{23}dz$	[N/m]	$x_{2}\;$= const.	(14f)

**Table 2 micromachines-11-00569-t002:** Stress-strain relations for fully elastic material behavior (E Young’s modulus, ν Poisson’s ratio).

Quantity	Strains ^1^		Stresses ^2^	
Normal components	$\varepsilon_{11}=\frac{1}{E}\left(\sigma_{11}-{\nu \sigma }_{22}\right)$	(22a)	$\sigma_{11}=\frac{E}{1-\nu^{2}}\left(\varepsilon_{11}+{\nu \varepsilon }_{22}\right)$.	(22b)
	$\varepsilon_{22}=\frac{1}{E}\left(\sigma_{22}-{\nu \sigma }_{11}\right)$	(23a)	$\sigma_{22}=\frac{E}{1-\nu^{2}}\left(\varepsilon_{22}+{\nu \varepsilon }_{11}\right)$.	(23b)
Shear components	$\gamma_{12}=\frac{2\left(1+\nu \right)}{E}\sigma_{12}$	(24a)	$\sigma_{12}=\frac{E}{2\left(1-\nu^{\;}\right)}\gamma_{12}$	(24b)

^1^ unitless, ^2^ unit (N/mm^2^).

**Table 3 micromachines-11-00569-t003:** Material and geometrical parameters of a silicon bending plate covered with hydrogel [7,25].

Parameter	Silicon Plate	Hydrogel Layer
Length a [mm]	4	variable
Width b [mm]	4	variable
Thickness $h_{1,2}$ [µm]	18	20
Young´s modulus $E_{1,2}$ [N/mm²]	131000	500 at 90%R.H.

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
