# Peer review of "Bistable Threshold Humidity Sensor Switch with Rectangular Bimorph Bending Plate"

_micromachines, 2020, doi:10.3390/mi11060569_

Round 1

Reviewer 1 Report

This manuscript reports the bistable threshold humidity sensor switch with rectangular bimorph bending plate. In contrast to previous bistable beam-like sensor with sheet metal and hydrogel, Si bimorph bending plate was demonstrated with hydrogel for humidity sensor. In addition, the experimental results were compared with a corresponding mechanical model. Although the subject has been extensively studied, more relevant and recent literature are recommended.

Below I have provided several remarks.

  1. Provide references: “Proper reference is requred regraig the statement of "This sensor shows a high sensitivity with regard to small changes and a corresponding closing (when swelling) or opening (when deswelling) of the microcontact." (line 41) and “Since SiO2 has a much lower coefficient of thermal expansion than silicon and is deposited at temperatures of several 100C” (line 69)
  2. Check equation 82
  3. What stands for [u.a.] in the Fig. 12? Is it arbitrary unit?
  4. In general, it is not recommended to use abbreviations in the abstract
  5. When you use an abbreviation in the text, define it in upon first use. 

Author Response

Dear reviewer,

We have received your helpful comments and appreciate your efforts. We followed your recommendation and inserted additional references. For better understanding, the design of Fig. 1 - 4 and Fig. 14 has been improved.

  1. Provide references: “Proper reference is requred regraig the statement of "This sensor shows a high sensitivity with regard to small changes and a corresponding closing (when swelling) or opening (when deswelling) of the microcontact." (line 41) and “Since SiO2 has a much lower coefficient of thermal expansion than silicon and is deposited at temperatures of several 100C” (line 69)

We insert the corresponding references for both items.

  1. Check equation 82

Eq. (82) has been corrected.

  1. What stands for [u.a.] in the Fig. 12? Is it arbitrary unit?

Yes, we meant “arbitary unit”. The abbreviation “a.u.” is been written out now.

  1. In general, it is not recommended to use abbreviations in the abstract

We have changed it properly.

  1. When you use an abbreviation in the text, define it in upon first use. 

The abbreviations have been inserted after the full name. See lines 13, 18, 19 and 33.

Reviewer 2 Report

This is a nice paper regarding the development and realization of the mechanical model. From this mechanical model with axial compression, the S-shape curve progression shows an increase in the width of the hysteresis which depends from the axial compression. And from the measured results the pre-deflected and partly with hydrogel covered Si-plate shows a snap through switching behavior. The paper is very well written, well organized and the flow of the material read very well. But I have some comment and questions:

1- References to other similar works are poor. You should consider a wider field and then focus on your work with recent references in this journal.

2- Why you choose the bimorph structure

3- Line 74 to 76 The goal of the following considerations is to derive analytical models for the description of rectangular bimorph plates with patterned layers that undergo a volume expansion (swelling) due to humidity as similar to the thermal expansion of any materials.

4- Table 2 equation 22b to σ11

5- Can you explain the transition from equation 67 to 68

6- Table 3 : does the young module of the hygrogel change with humidity. In this case you must keep account in the calculation

7- Line 422: equation 70 need some explanation

8- Line 481: how you directly introduced the relative humidity in figure 10 without giving any indication in advance.

9- Line 493: Inserting Eqs. (81) and (81) into Eq. (82)

10- Figure 12 is to be explained further by introducing the effect of relative humidity

11- the data series to be removed in figure 12

12- line 511 and line 404 the coefficient thermal expansion of SiO2 is 5,6 10-7K,

13- How to control the thickness of deposition of the hydrogel layer and how this thickness acts on the bimorph structure.

14- Line 584: the extraction of φ0 = 0.57° from Fig. 13 is it right

15- Figure 12 to be corrected and it would be nice to show the results for φ0 = 0.57°

16- Why you add the equation 83 just before the conclusion

17- In perspective what is you recommendation for energy harvesting

18- Does the weight of the dust influence the deviation in the deflection amplitude

19- The experimental part is very well explained which misses on the other hand the discussion of the results.

Author Response

Dear reviewer,

We have received your helpful comments and appreciate your efforts. We follow your recommendation and insert literature. For better understanding, the design of Fig. 1 - 4 and Fig. 14 has been improved.

    1. References to other similar works are poor.

We inserted additional references.

  1. Why you choose the bimorph structure

We wanted to separate the humidity-sensitive area of the sensor switch from the electrical components (micro contacts) by using a plate instead of the previously used beam. The bimorph effect ensures a sufficiently large deflection of the bending plate out of plane und, in the same way, enables to apply MEMS fabrication technologies.

  1. Line 74 to 76 The goal of the following considerations is to derive analytical models for the description of rectangular bimorph plates with patterned layers that undergo a volume expansion (swelling) due to humidity as similar to the thermal expansion of any materials.

Yes, this is our goal.

  1. Table 2 equation 22b to σ11

The numbering of the equations was corrected.

  1. Can you explain the transition from equation 67 to 68.

The transition from equation 67 to 68 is now explained in lines 388-397.

  1. Table 3 : does the young module of the hygrogel change with humidity. In this case you must keep account in the calculation

Yes, you are right: The Young's modulus is changing with humidity. However, to keep the model simple, we considered here in this paper a constant Young's modulus: In this model only the hydrogel expansion is considered. In future work, the change of the Young's modulus will be considered to get better simulation results.

  1. Line 422: equation 70 need some explanation

An explanation of equation 70 can now be found in line 446.

  1. Line 481: how you directly introduced the relative humidity in figure 10 without giving any indication in advance.

For better understanding, line 507-510 now explains in detail the relationship betwee humidity and deflection angle, as shown in Fig. 10.

  1. Line 493: Inserting Eqs. (81) and (81) into Eq. (82)

This error has been corrected. You can find the improvement in line 519-520.

  1. Figure 12 is to be explained further by introducing the effect of relative humidity

We have provided this explanation now in lines 530-532.

   11- the data series to be removed in figure 12

Is done now accordingly.

    12- line 511 and line 404 the coefficient thermal expansion of SiO2 is 5,6 10-7K,

This error has been corrected. You will now find the correction in line 426 and line 545.

  1. How to control the thickness of deposition of the hydrogel layer and how this thickness acts on the bimorph structure.

Line 643-645.

To produce the hydrogel layer, the cavity is filled up with a polymer solution to the upper chip edge. Since the polymer solution has the same polymer concentration and only the coating area is varied via the silicone stamp diameter, the layer thickness is reproducible (Fig. 14).

  1. Line 584: the extraction of φ0 = 0.57° from Fig. 13 is it right

This error was corrected. You will find the correction in line 606 and 609.

  1. Figure 12 to be corrected and it would be nice to show the results for φ0 = 0.57°

The angle φ0=0.00057 was inserted and the corresponding deflection was calculation. It has an influence on the calculated deflection. Fig. 14 shows the calculated deflection compared to the measured deflection.

  1. Why you add the equation 83 just before the conclusion

Eq. (83) is now included in the text now at the right place (line 613).

  1. In perspective what is you recommendation for energy harvesting

The dynamic behavior of the sensor switch is too slow for energy harvesting but suitable for the switching process, if the snap-through process is achieved.

  1. Does the weight of the dust influence the deviation in the deflection amplitude

We could not observe any influence of the dust weight. The pressure sensor was supplied with voltage and the output signal was measured. The sensor was tilted by 90 degrees and the output signal was measured again. No signal change could be observed. During longer operation times a deposit of dust particles on the surface cannot be excluded. The contamination of the sensor by dust can be counteracted by a suitable design or an adequate encapsulation. However, this is not part of this work and must be investigated in long-term application.

  1. The experimental part is very well explained which misses on the other hand the discussion of the results

We have included now the discussion of the results in the summery.

Round 2

Reviewer 2 Report

Response after review

  • The author has answered all the asked questions and all comments are taken into consideration,
  • The changes made in the text and figures make the article more understandable.
  • I recommend its publication after correction in figure 12